# In Vivo Selection of S/MAR Sequences to Favour AAV Episomal Maintenance in Dividing Cells

**DOI:** 10.3390/ijms252312734

**Published:** 2024-11-27

**Authors:** Andrea Llanos-Ardaiz, Aquilino Lantero, Leire Neri, Itsaso Mauleón, Marina Ruiz de Galarreta, Laia Trigueros-Motos, Nicholas D. Weber, Veronica Ferrer, Rafael Aldabe, Gloria Gonzalez-Aseguinolaza

**Affiliations:** 1Vivet Therapeutics S.L., 31008 Pamplona, Spain; allanos@vivet-therapeutics.com (A.L.-A.); old-alantero@vivet-therapeutics.com (A.L.); lneri@vivet-therapeutics.com (L.N.); mrdegalarreta@vivet-therapeutics.com (M.R.d.G.); ltrigueros@vivet-therapeutics.com (L.T.-M.); nweber@vivet-therapeutics.com (N.D.W.); 2DNA & RNA Medicine Division, Centre for Applied Medical Research (CIMA), University of Navarra, 31009 Pamplona, Spain; imauleon@unav.es; 3Vivet Therapeutics S.A.S., 75008 Paris, France

**Keywords:** S/MAR, AAV, episomal maintenance, replication, NMD, liver, gene therapy

## Abstract

Adeno-associated viral (AAV) vector-mediated gene therapy has emerged as a promising alternative to liver transplantation for monogenic metabolic hepatic diseases. AAVs are non-integrative vectors that are maintained primarily as episomes in quiescent cells like adult hepatocytes. This quality, while advantageous from a safety perspective due to a decreased risk of insertional mutagenesis, becomes a disadvantage when treating dividing cells, as it inevitably leads to the loss of the therapeutic genome. This is a challenge for the treatment of hereditary liver diseases that manifest in childhood. One potential approach to avoid vector genome loss involves putting scaffold/matrix attachment regions (S/MARs) into the recombinant AAV (rAAV) genome to facilitate its replication together with the cellular genome. We found that the administration of AAVs carrying the human β-interferon S/MAR sequence to neonatal and infant mice resulted in the maintenance of higher levels of viral genomes. However, we also observed that its inclusion at the 3′ end of the mRNA negatively impacted its stability, leading to reduced mRNA and protein levels. This effect can be partially attenuated by incorporating nonsense-mediated decay (NMD)-inhibitory sequences into the S/MAR containing rAAV genome, whose introduction may aid in the development of more efficient and longer-lasting gene therapy rAAV vectors.

## 1. Introduction

Recombinant adeno-associated viruses (rAAVs) have emerged as one of the most promising tools for gene therapy [1]. Their unique properties, together with advances in their development, have positioned them as a leading choice for the treatment of genetic diseases, including those affecting the liver [2]. The selection and modification of AAV serotypes, including through protein engineering and chemical modifications, has allowed the design of AAVs to have a natural affinity for specific tissues such as the liver, ensuring the efficient delivery and expression of the therapeutic gene in the target tissue [3]. One important property of rAAV vectors is that they can mediate long-term gene expression due to their ability to persist in cells as episomes for extended periods, as has been reported in numerous preclinical studies and, importantly, in clinical trials [4]. This holds true for non-dividing or slowly dividing cells like adult quiescent hepatocytes. However, in dividing cells, such as the hepatocytes in a growing or heavily damaged liver, rAAV genomes are rapidly lost [5,6]. This represents a major drawback of using rAAVs for the treatment of liver diseases in infancy or in conditions characterised by high hepatocyte turnover.

One potential strategy to overcome this limitation could be repeated administrations of the therapeutic vector. Currently, this option involves a number of complexities due to the strong inhibitory effect of the neutralising antibodies (NAbs) developed after the first vector injection [7]. Although strategies like mechanical or enzymatic elimination of NAbs are under evaluation, their applicability will be most likely restricted to patients with low NAb titres [8,9]. Research has also been dedicated to using different capsid serotypes for subsequent administrations [10,11,12] and to inducing immunological tolerance to AAVs with the initial treatments [13,14,15]. Despite these efforts, a simpler solution could be to engineer the AAV genome, providing it with the capacity to replicate together with the cell and thus prevent the necessity of re-administration.

Scaffold/matrix attachment regions (S/MARs) are specific DNA sequences that anchor chromatin to the nuclear scaffold or matrix. They are specialised DNA elements that have been found in all eukaryotes [16]. These sequences play a crucial role in the structural organisation of the genome and the regulation of gene expression. Additionally, they are essential for various nuclear processes, including DNA replication, transcription, and repair [17]. S/MAR sequences do not exhibit sequence conservation, yet their secondary structure features seem to be conserved and play a crucial role in their function. These sequences are DNA fragments of 300–3000 bp in length that contain characteristics such as origins of replication (OriC), A + T rich sequence motifs, MAR signatures, and Topoisomerase-II binding sites [18]. Several methods have been developed to identify S/MAR sequences [18]; however, according to Evans et al., none of them have significant predictive power [19]. Among the S/MAR sequences that have been described so far, the one corresponding to the 5′ region of the human β-interferon gene (hIFN-S/MAR) has been extensively characterised and used to generate replication-competent plasmids [20,21,22]. hIFN-S/MAR has been shown to provide mitotic stability to episomal plasmids that divide once per cell cycle and are transmitted over many cell generations in culture without selection [23,24]. Mechanistic studies showed that interaction of the S/MAR with the nuclear matrix is required [25] and that transcription must go through the S/MAR element for it to be functional [21,26].

It has been shown that administration by hydrodynamic injection or the use of nanoparticles to deliver hIFN-S/MAR-containing plasmids with different transgenes to the livers of mice can result in long-term protein expression. This contrasts with the transient expression observed with the plasmids lacking S/MAR [27,28,29]. Additionally, its functionality has also been tested in other organs such as the eye [30,31] and in the context of other gene delivery vectors like integration-deficient lentiviral vectors [32,33]. However, the incorporation of S/MAR elements in the context of AAV vectors has been tested only in cell culture [34], in which proliferating HeLa cells transduced with an AAV-S/MAR, and following mild antibiotic selection, were able to establish colonies that maintained long-term transgene expression in the absence of further selection or vector integration.

In the present work, we introduced various S/MAR sequences into the AAV genome and evaluated their activity to improve AAV genome maintenance in vivo in proliferating hepatocytes. After selecting the best-performing sequence in neonatal mice, we conducted further studies across animals of different ages. Our data revealed that the hIFN-S/MAR, in forward orientation and upstream of the polyA sequence, was the most effective, and consistently increased the number of vector genomes at least twofold in the different experimental conditions. However, we also found that the inclusion of the S/MAR sequence upstream of the polyA significantly impairs transgene expression, which we found can be mitigated by the inclusion of nonsense-mediated decay (NMD)-inhibitory sequences.

## 2. Results

### 2.1. Selection of S/MAR Sequences

We first selected several previously identified S/MAR sequences and inserted them into an AAV expression cassette to determine their impact on AAV genome maintenance when administered to neonatal mice. Specifically, we placed five different S/MAR elements—interferon-β (IFN-β), dihydrofolate reductase (DHFR), cold shock protein β (CSP-β), c-Myc, and Kaposi’s sarcoma-associated virus (KSHV)—in both forward (F) and reverse (R) orientations into an AAV expression cassette between an open reading frame (ORF) for enhanced green fluorescent protein (*eGFP*) and the poly-A sequence (Figure 1A, Appendix A Table A1). *eGFP* expression was under the transcriptional control of the liver-specific alpha-1-antitrypsin (AAT) promoter. We produced 10 AAV-S/MAR vectors with the hepatotropic Anc-80 serotype alongside a control virus without any S/MAR element (ssAAV-Anc80-eGFP) and administered them intravenously to groups of 6–8 C57BL/6 mice 24 h after birth at a dose of 5 × 10^12^ viral genomes per kilogram of body weight (vg/kg).

Mice were sacrificed at 1, 3, and 6 weeks post-inoculation, and liver samples were collected to assess the S/MAR sequence impact on AAV genome maintenance in a highly proliferating host organ (Figure 1B). Quantification of viral genomes revealed that the IFN-β forward (IFN-β-F) and the c-Myc reverse (c-Myc-R) groups had more viral genomes at 3 and 6 weeks post-inoculation than the other nine groups, including those inoculated with the ssAAV-Anc80-eGFP control vector (Figure 1C). This increase was statistically significant only for the c-Myc-R group.

Therefore, the assessment of S/MAR sequences on AAV vector maintenance revealed that IFN-β-F and c-Myc-R positively impacted AAV genome maintenance following neonatal administration.

### 2.2. Efficacy of S/MAR Vectors in Mice at Different Infantile/Juvenile Ages

Administration of AAV to neonates resulted in a significant loss of AAV-transduced cells from week 1 to week 6 post-administration (Appendix A, Figure A1), a decline that did not correlate with the hepatocytes’ proliferation rate, suggesting that additional factors are likely contributing to the reduction in genomes. Thus, we decided to evaluate the activity of IFN-β-F and c-Myc-R in vectors administered later in life, when the rate of hepatocyte proliferation is lower. AAV vectors were intravenously administered to C57BL/6 mice at a dose of 5 × 10^12^ vg/kg. Control and IFN-β-F vectors were administered to mice at 2, 3, and 4 weeks of age, and c-Myc-R was administered to mice at 2 and 3 weeks of age only (Figure 2A). Mice were sacrificed 4 weeks (2-week-old cohort) and 3 weeks (3- and 4-week-old cohorts) post-injection, and liver samples were collected for evaluation (Figure 2B). 

Assessment of AAV genomes in hepatic samples 3–4 weeks after treatment showed that mice treated with the AAV-IFN-β-F vector had a higher number of viral genomes compared with mice treated with the control and c-Myc-R vectors. This effect was more pronounced when the AAV was inoculated into 3-week-old mice vs. 2-week-old mice (10-fold vs. 2-fold increase) (Figure 2C), and no differences were observed in the animals injected at 4 weeks. Additionally, three-week-old treated mice showed higher transgene expression than control mice when the IFN-β-F element was present, although the improvement in transgene expression was less significant than the impact on AAV genome maintenance. Interestingly, the presence of the c-Myc-R sequence reduced eGFP expression, particularly in three-week-old treated mice, despite presenting similar AAV genome levels as the control mice (Figure 2D).

A comparison of the *eGFP* mRNA expression per AAV genome detected across the three treatment groups (Figure 2E) revealed a negative impact of the IFN-β-F sequence compared with the control vector. Evaluation of the percentage of eGFP-positive hepatocytes supported the results of vector genomes and *eGFP* mRNA (Figure 2F), showing no differences between the three groups in two-week-old administered mice. In addition, the IFN-β-F element (vs. control, *p* < 0.001) presented a positive impact and the c-Myc-R sequence promoted a negative effect in three-week-old administered mice (vs. control, *p* < 0.001).

### 2.3. The Inclusion of the S/MAR Sequence Resulted in a Reduction in mRNA Stability

The inclusion of the S/MAR sequence between the ORF and the polyA sequence results in an increase in the maintenance of rAAV viral genomes in dividing cells (Figure 3A). However, it also produces an mRNA with an extended 3′ UTR that could have an impact on mRNA stability. To investigate mRNA stability, we employed two cDNA synthesis methods: one using random primers to reverse-transcribe all RNA molecules, and another using oligo-dT primers to amplify only complete mRNAs with a polyA tail. When comparing the proportion of full-length *eGFP* cDNA molecules to total *eGFP* mRNA molecules in samples from mice treated at 3 weeks of age and sacrificed 6 weeks later, we found that a smaller proportion of full-length molecules were present in the S/MAR group (Figure 3B). This suggests that the extended 3′ UTR associated with the inclusion of the S/MAR sequence reduced *eGFP* mRNA stability, thereby diminishing the potency of the expression cassette.

These results showed that the IFN-β-F S/MAR sequence reduced the AAV genome loss associated with hepatocyte proliferation, with the strongest effect observed when AAVs were administered to three-week-old mice. However, the positive effect of the S/MAR sequence on AAV genome maintenance was offset by the negative impact on transgene expression and stability, limiting its potential therapeutic benefit.

### 2.4. Effect of NMD-Inhibitory Sequences on eGFP-S/MAR mRNA Stability

A potential explanation for the lower number of full-length *eGFP* mRNA copies in the S/MAR group is that the mRNA molecules with long 3′ UTRs are susceptible to degradation via the nonsense-mediated decay (NMD) pathway. Several studies have identified mRNAs with long 3′ UTRs that escape NMD degradation due to the presence of specific sequences that block NMD recognition [35,36,37]. Based on this information, we selected four NMD-inhibitory element sequences—derived from two human genes (hTRAM and hTMED [38]) and two viral genes (RSV-RSE [39] and TCV [40])—to evaluate their potential to improve mRNA stability of sequences containing S/MAR in the 3′ UTR.

The four sequences (Appendix A Table A2)—RSV-RSE, hTMED, TCV, and hTRAM—were placed between the transgene ORF and the S/MAR element. We produced 10 AAV vectors containing the AAT promoter, with or without the IFN-β-F S/MAR element and with or without any of the four NMD-inhibitory sequences (Figure 4A). These vectors were administered intravenously to 3-week-old mice at a dose of 5 × 10^12^ vg/kg, and 6 weeks later, the mice were sacrificed and liver samples analysed as previously described.

Consistent with the previous findings, mice treated with the control S/MAR vector exhibited more than a fivefold increase in viral genomes compared with the control vector without the S/MAR element. All vectors with an NMD-inhibitory sequence, but without the S/MAR element, had a similar number of viral genomes compared with the control vector. However, the inclusion of the S/MAR sequence with any NMD-inhibitory sequence resulted in a twofold reduction in vector genomes compared with the same NMD vector without the S/MAR sequence. These results suggest that the NMD-inhibitory elements interfere with the S/MAR element’s role in rAAV genome maintenance (Figure 4C).

Analysis of *eGFP* transgene mRNA expression indicated that all vectors without the S/MAR element produced similar *eGFP* mRNA levels except for AAT-RSV, which showed a significant increase (Figure 4D). In line with the previous results, once again we found that the introduction of the S/MAR element reduced the mRNA levels per vector genome (Figure 4E). Interestingly, the inclusion of three NMD-inhibitory sequences (RSV-RSE, TCV, and hTRAM) in the S/MAR-containing vectors improved the transgene mRNA expression per genome, reaching levels comparable to those observed in mice treated with the AAT-eGFP control vector. Additionally, in the vectors lacking the S/MAR sequence, the addition of two of the NMD-inhibitory elements (RSV and TCV) positively affected transgene mRNA expression per genome in comparison with the AAT-eGFP control vector that lacked the S/MAR element. These findings suggest that NMD-inhibitory elements generally restore the relative mRNA expression of vectors containing the S/MAR sequence in the mRNA 3′ UTR. However, they interfere with the capacity of the S/MAR sequence to prolong the presence of AAV genomes in a growing liver.

Some studies have suggested a link between transcriptional potency and S/MAR activity, i.e., increasing the promoter potency could enhance S/MAR activity [41]. Given our observation that NMD-inhibitory sequences positively influence the stability of *eGFP* mRNAs with a long 3′ UTR, we investigated the effects of using a strong promoter by adding the albumin enhancer (EnhAlb) sequence to the AAT promoter (EnhAlbAAT). We excluded the hTRAM NMD-inhibitory element, which was found to have a strong negative impact on AAV genome stability (Figure 4C).

We repeated the experiment as before, with vectors containing the EnhAlbAAT promoter and the three most promising NMD sequences (RSV-RSE, TCV, hTMED) (Figure 4B and Figure 5A). Six weeks after virus administration, all groups of mice treated with vectors lacking the S/MAR element exhibited statistically similar numbers of viral genomes. In contrast, mice treated with vectors containing both the NMD-inhibitory sequences and the S/MAR element showed a 5- to 10-fold increase in viral genomes vs. the EnhAlbAAT-eGFP control vector and a 1- to 2-fold increase vs. the S/MAR vector without any NMD-inhibitory sequence. However, this increase in viral genome count vs. the control vector was statistically significant only in mice treated with the EnhAlbAAT-RSV-S/MAR and Enh AlbAAT-hTMED-S/MAR vectors (Figure 5B).

*eGFP* mRNA expression analysis revealed similar levels across all groups, except in animals treated with the vector containing both the S/MAR and hTMED NMD-inhibitory elements, which exhibited higher mRNA expression than either control group, EnhAlbAAT-eGFP and EnhAlbAAT-S/MAR (Figure 5C), as did the vector containing hTMED without S/MAR, which had comparable but slightly lower levels than the vector with S/MAR. The S/MAR element may interfere with *eGFP* mRNA stability, as 5- to 8-fold lower mRNA levels per genome were seen compared with the same vectors without the S/MAR sequence. This negative impact was only partially mitigated by the hTMED NMD-inhibitory element (Figure 5D).

Interestingly, when eGFP protein expression was assessed by Western blot of the hepatic samples (Appendix A Figure A2), hTMED significantly increased eGFP protein levels, regardless of the presence of the S/MAR element in the vector, compared with the corresponding control vectors, EnhAlbAAT-eGFP and EnhAlbAAT-S/MAR (Figure 5E). Finally, the assessment of the eGFP-positive area in liver sections showed a slight increase (1.4-fold) in the number of eGFP-positive hepatocytes when the NMD-inhibitory elements were included vs. either of the control vectors that lacked any NMD-inhibitory element (Figure 5F).

These findings indicate an interplay between the S/MAR sequence and NMD-inhibitory elements, with effects on AAV genome maintenance and mRNA stability, respectively. Achieving a balance between these elements will be crucial for optimising transgene expression while minimising AAV genome loss in growing livers.

## 3. Discussion

Neonatal gene transfer offers several advantages, including achieving therapeutic effects before disease onset, requiring lower vector quantities, allowing for a high vector-to-cell ratio, and operating within a relatively immature immune system [42]. However, one of the main limitations for the therapeutic use of AAV vectors for the treatment of genetic disorders at an early age is that due to their episomal nature, the genomes are lost upon cell division [2,43]. The mechanism that is generally accepted to explain this loss is a simple dilution effect during hepatocyte turnover. However, other mechanisms may also be at play, including AAV ssDNA and linear dsDNA genome degradation via DNA repair machinery [44], which is more highly active in proliferating cells. Even in organs with low cell turnover, AAV genomes are gradually lost over time, likely necessitating re-administration, which remains challenging due to the neutralising immune response [45,46]. Thus, the development of AAV genomes that can replicate with cell division would be of extraordinary value. Several groups have shown that the introduction of S/MAR elements in plasmids or episomal recombinant viruses confer to these DNA sequences the capacity to replicate together with the cellular genome [24,32,47]. If this capability could be extended to AAV vectors, it could offer a way for the obstacles associated with the loss over time of the vector episomes to be overcome.

As a first step, we tested AAVs containing different S/MAR sequences, in forward and reverse orientations, in the highly proliferative liver present in neonatal mice. It was found that the presence in the viral vector of hIFN-β forward and c-Myc reverse reduced the magnitude of viral genome loss 3 and 6 weeks after vector injection in comparison with a vector without S/MAR. Interestingly, the inclusion of the same S/MAR sequence in the opposite orientation had no effect, nor did the inclusion of the other S/MAR sequences tested. The functionality of the hIFN-S/MAR has been previously reported in several studies in cell culture, primary cells derived from blood and in vivo in the liver or the eye [27,30,47,48,49,50]. In most studies, an initial step using selection via antibiotic resistance has been employed to increase the proportion of cells that establish a durable expressing episome containing the S/MAR element. During and after this first establishment phase, the host cell’s replication machinery can be utilised. S/MAR episomes then replicate once during the early S phase of the cell cycle in a manner regulated in concordance with the replication of the cellular genome [51].

However, when S/MAR vectors are tested in highly proliferative systems in the absence of positive selection that favours the early establishment of episomes, such as in zebrafish [52] and in neonate mice, the AAV genomes are rapidly lost. To address this high rate of proliferation, the two selected S/MAR sequences were again tested in mice at different ages (2, 3, and 4 weeks) where hepatocytes are still proliferating but at lower rates. Under these conditions, we found that the hIFN-S/MAR sequence provided a proliferative advantage for the AAV genomes, particularly at 3 weeks of age, consistent with the results seen when this sequence has been used in other vectors [32,33,34,53,54,55,56]. Interestingly, this positive effect of the hIFN-S/MAR sequence was lost when the animals were injected at 4 weeks of age, coinciding with a lack of or reduced hepatocyte proliferation. This positive effect was also less significant when mice were injected at 2 weeks of age, coinciding with a higher hepatocyte proliferation rate. Thus, the S/MAR sequence demonstrates a positive impact when AAV genome dilution occurs due to hepatocyte proliferation, but only when there is a low to moderate proliferation rate, as in the hepatocytes of 3-week-old mice. Thus, there is a specific window where the conditions for S/MAR functionality are the most ideal. Interestingly, these results correlated with a significantly higher percentage of eGFP-positive hepatocytes in the AAV-hIFN-S/MAR than in the control group. Intriguingly, the tenfold advantage that we see in terms of the genome is not translated into higher mRNA levels or percentages of transduced hepatocytes, which were improved only twofold; in fact, the mRNA/genome ratio is lower with the inclusion of the hIFN-S/MAR sequence in comparison with the control in all the different treatment age groups. Therefore, the hIFN-S/MAR sequence promotes the maintenance of AAV episomes in a greater proportion of hepatocytes, as suggested by the quantification of AAV genomes and eGFP-positive hepatocytes.

These observations indicate that the presence of the S/MAR sequence in the messenger RNA may be detrimental. This finding contrasts with previous reports, which indicate that S/MAR sequences enhance transgene expression [57]. However, the positive effect was observed for integrated vectors rather than for episomes and depended on mechanisms involving transcriptional regulation. Several studies have assessed the impact of different promoters on S/MAR sequence activity [58]. However, the impact that S/MAR sequences residing within the transcriptional unit have on transcript stability or translation has not been evaluated. It is known only that the S/MAR sequences must be transcribed to exert their nuclear retention function in episomal vectors, but this could be due to questions of DNA accessibility and not have any bearing on RNA.

The placement of the S/MAR sequence within the expression cassette produces mRNA transcripts with a longer 3′ UTR, which could impact mRNA stability, as suggested by the observed reduced levels of mRNAs containing the S/MAR sequence. One hypothesised mechanism for this decrease in the stability of transcripts with longer 3′ UTR is nonsense-mediated decay (NMD). NMD is a crucial RNA quality control and gene regulation mechanism conserved across eukaryotes. Its primary purpose is to degrade mutant mRNAs containing premature termination codons (PTCs), thus preventing their translation. mRNA molecules with long 3′ UTRs, either resulting from PTCs or artificially extended 3′ UTRs, can trigger NMD by enhancing up-frameshift 1 (UPF1) assembly on the targeted mRNAs [59,60,61]. While many endogenous human and some viral mRNAs possess long 3′ UTRs, they are resistant to NMD [62] via inhibitory elements of NMD activity in cis, which function to increase mRNA stability and promote NMD evasion when inserted into a heterologous NMD-sensitive mRNA by interfering with binding of the UPF complex [38].

Among the four NMD-inhibitory sequences that we evaluated, three significantly improved mRNA expression per genome present in the cells, suggesting a beneficial effect of these sequences on mRNAs containing the S/MAR sequence. This is consistent with observed results when these elements have been placed in mRNAs with long 3′ UTRs that are degraded by the NMD machinery [38,40]. However, there appears to be a competitive interplay between the NMD-inhibitory sequences and the S/MAR sequence, as the NMD-inhibitory sequences can block S/MAR activity, a phenomenon that is bypassed when a stronger promoter is used. This discrepancy in the interference between the two elements according to the promoter sequence within the vector does not support the hypothesis of a robust mechanistic competition between the two pathways. Several studies have shown that transcriptional activity is required for S/MAR sequence functionality; therefore, a stronger promoter should enhance S/MAR sequence efficacy and increase AAV genome maintenance even with the inclusion of NMD-inhibitory sequences. However, this appears to reduce the positive effect these sequences have on mRNA stability in transcripts with long 3′ UTRs. Interestingly, one of the NMD-inhibitory sequences, hTMED, positively impacted mRNA translation, significantly improving transgene production and resulting in a more potent vector.

A more detailed analysis of the interplay between these sequences, the S/MAR sequence, the NMD-inhibitory sequence, and the promoter will be necessary to develop an AAV vector candidate that best improves genome maintenance in a proliferating liver while mitigating the negative impact of long 3′ UTRs on mRNA stability.

The primary limitation of incorporating additional elements into the AAV genome is the significant reduction in the cloning capacity of recombinant vectors. This limitation restricts the number of diseases that can benefit from these vectors to those caused by mutations in relatively small genes, such as tyrosinemia, phenylketonuria, and citrullinemia. Strategies to overcome this limitation are currently being explored, including the identification of smaller S/MAR sequences and the delivery of dual vectors.

We have identified an S/MAR candidate that functions to improve AAV genome maintenance in proliferating hepatocytes, but the most favourable scenario for its most optimal application still needs to be elucidated fully. Additionally, the inclusion of NMD-inhibitory sequences has shown concrete benefits in mRNA translation, but the nuances of including both sequence elements and the nature of the transcriptional promoter needs further investigation. These and further future advances will ideally open the door for expanding the potential patient pool for AAV-mediated gene therapy to include very young children.

## 4. Materials and Methods

### 4.1. Cloning and rAAV Vectors Construction

#### 4.1.1. Cloning of Plasmids for S/MAR Sequence Selection

The different S/MAR sequences were either purchased as custom-order plasmids from ThermoFisher (ApoB, CSPi, c-Myc, and KSHV) or PCR amplified from the pEPI-IR plasmid (DHFR and IFN-β) [22,63] that was kindly provided by Ruben Hernández’s lab.

For the cloning of the pAAV-AAT-EGFP-IFN-β-R-pA plasmid, specific primers were generated for the amplification of the inverted IFN-β sequence from the pEPI-IR plasmid and its introduction into the pAAV AAT eGFP polyA plasmid that was previously digested with EcoRI (#R3101, New England Biolabs, Ipswich, MA, USA) using In-Fusion HD cloning kit (In Fusion HD Cloning Plus, 638911, Clontech, Mountain View, CA, USA), following manufacturers’ instructions.

S/MAR INV. Forward: CGGCCGCGACTCTAGAATTCGAATTCTATCAAATATTTAAS/MAR INV. Reverse: GAAGGCACAGGAATTCAGATCTAAATAAACTTATAA

For the cloning of the pAAV-AAT-EGFP-IFN-β-F-pA plasmid, the IFN-β sequence was enzymatically extracted, using EcoRI restriction enzyme, from the pEPI-IR plasmid and introduced into the pAAV AAT eGFP polyA plasmid that was previously digested with EcoRI and dephosphorylated using the T4 ligase (#M0202T, New England Biolabs, Ipswich, MA, USA).

For the cloning of the pAAV-AAT-EGFP-DHFR-R-pA, specific primers were generated for the introduction of the inverted DHFR sequence from the pEPI-IR plasmid into the pAAV AAT eGFP polyA plasmid that was previously digested with EcoRI using In-Fusion HD cloning kit.

DHFR INV. Forward: 5′-CGGCCGCGACTCTAGAATTCAGAGGCATCAGGTAGTGAGT-3′DHFR INV. Reverse: 5′-GAAGGCACAGGAATTCTGTGAAGAGACACCATGACC-3′

For the cloning of the pAAV-AAT-EGFP-DHFR-F-pA plasmid, the DHFR sequence was enzymatically extracted using EcoRI restriction enzyme from the pEPI-IR plasmid and introduced into the pAAV AAT eGFP polyA plasmid that was previously digested with EcoRI and dephosphorylated using the T4 ligase.

For the cloning of pAAV-AAT-EGFP-CSP-F-pA and pAAV-AAT-EGFP-CSP-R-pA, the hCSP-B-MAR sequence was enzymatically restricted from its containing PMK-RQ vector (GeneArt, ThermoFisher Scientific, Waltham, MA, USA) and introduced into the pAAV AAT eGFP polyA plasmid that was previously digested with EcoRI and dephosphorylated using the T4 ligase. Clones were checked via enzymatic digestion to determine hCSP-B-MAR orientation.

For the cloning of pAAV-AAT-EGFP-KSHV-F-pA and pAAV-AAT-EGFP-KSHV-R-pA, the KSHV-ARE sequence was enzymatically restricted from its containing PMAT-T vector (GeneArt, ThermoFisher Scientific) using Pvu1 and EcoRI and introduced into the pAAV AAT eGFP polyA plasmid that was previously digested with EcoRI and dephosphorylated using the T4 ligase. Clones were checked via enzymatic digestion to determine KSHV orientation.

For the cloning of pAAV-AAT-EGFP-APOB-F-pA and pAAV-AAT-EGFP-APOB-R-pA, the APOB sequence was enzymatically restricted from its PMK-containing vector (GeneArt, ThermoFisher Scientific) using EcoRI and introduced into the pAAV AAT eGFP polyA plasmid that was previously digested with EcoRI and dephosphorylated using the T4 ligase. Clones were checked via enzymatic digestion to determine APOB orientation.

For the cloning of pAAV-AAT-EGFP-CMYC-F-pA and pAAV-AAT-EGFP-CMYC-R-pA, the CMYC sequence was enzymatically restricted from its PMK-RQ-containing vector (GeneArt, ThermoFisher Scientific) using EcoRI and introduced into the pAAV AAT eGFP polyA plasmid that was previously digested with EcoRI and dephosphorylated using the T4 ligase. Clones were checked via enzymatic digestion to determine CMYC orientation.

#### 4.1.2. Cloning of NMD Inhibitor Sequences Containing Plasmids

Four different NMD-inhibitory sequences were custom ordered from GenScript Biotech Corporation, Piscateville, NJ, USA: RSVE-RSE, hTMED, TCV, and hTRAM.

For the construction of the pAAV-AAT-eGFP-NMDs-pA, pAAV-AAT-eGFP-NMDs-S/MAR-pA, pAAV-EnhAlbAAT-eGFP-NMDs-pA, and pAAV-EnhAlbAAT-eGFP-NMDs-S/MAR-pA vectors, four starting plasmids previously cloned in-house, pAAV-AAT-eGFP-pA, pAAV-AAT-eGFP-IFN-pA, pAAV-EnhAlbAAT-eGFP-pA, and pAAV-EnhAlbAAT-eGFP-IFN-pA, respectively, were used. These plasmids were linearised by digestion with XbaI (#R0145. New England Biolabs). RSV-RSE, hTMED, TCV, and hTRAM NMD-inhibitor sequences were amplified by PCR using the following:RSV-RSE Forward 5′-CGGCCGCGACTCTAGGGAGGGCCACTGTTCTCACTG-3′RSV-RSE Reverse 5′-GCACAGGAATTCTAGTTCCGGATCACGAAGACAGGTG-3′hTMED Forward 5′-CGGCCGCGACTCTAGAAAGCCTCTTCCTGATGATCCC-3′hTMED Reverse 5′-GCACAGGAATTCTAGAATTATGCCAATCAACTGTCAAATT-3′TCV Forward 5′-CGGCCGCGACTCTAGTACGGTAATAGTGTAGTCTTCTCA-3′TCV Reverse 5′-GCACAGGAATTCTAGGGGCAGGCCCCCCCCCCG-3′hTRAM Forward 5′-CGGCCGCGACTCTAGTGAATTATAAACTAATTGATTAATG-3′hTRAM Reverse 5′-GCACAGGAATTCTAGTAGTCTAAGCTAAAATATTTTTTTC-3′

In-Fussion kit was used according to manufacturer’s instructions to mix the linearised starting plasmids and their respective inserts with the NMDs sequences.

#### 4.1.3. Cloning of IFN-β S/MARs in Different Positions

To generate the pAAV-AAT-eGFP-pA-IFN-β-F vector, the “AAT-eGFP-pA” insert was PCR-amplified using specific primers and introduced in the pAAV-IFN vector, previously linearised with Mlu1(#R0198, New England Biolabs), using the In-Fusion HD Cloning Plus kit.
AAV-AAT-Forward 5′-CTGCGGCCGCACGCGTGGTGCCACCCCCTCCACCTT-3′IFN-pA-Reverse 5′-AAGTTTATTTAGATCACGCGTCCATAGAGCCCACCGCATCC-3′

To generate the pAAV-IFN-β-F-AAT-eGFP-pA vector, two starting plasmids obtained from CIMA, pAAV-eGFP-pA and pAAV-IFN-β-For, were used. The final vector was obtained following digestion with EcoR1 of pAAV-IFN-β-For and with Mlu1 of pAAV-GFP-pA. Both linearised vectors were then joined by ligation using the T4 ligase.

### 4.2. AAV Production

rAAV serotype Anc80 vectors with AAV2 ITRs were produced by polyethyleneimine-mediated co-transfection in HEK-293T cells (ATCC^®^ CRL-3216™, Manassas, VA, USA). HEK-293Ts were cultured in 150 mm plates in Dulbecco’s modified Eagle’s medium with pyruvate (DMEM. #11995073, GIBCO, Miami, FL, USA) supplemented with 10% heat-inactivated foetal bovine serum (FBS, #10270-106, GIBCO), 1% penicillin and streptomycin (#15140-122, GIBCO) and incubated at 37 °C and 5% CO2 for 24 h. Cells were then transfected with a plasmid mix, including pδF6 (previously generated at CIMA), pkAnc80AAP2 (Lab. Luk H. Vandenberghe (Boston, MA, USA)), and a plasmid including the expression cassette for each construction mixed in a 3:1 volume ratio with polyethyleneimine. The cells were harvested 72 h later, and viral particles were purified by iodixanol (OptiPrep^TM^ #D1556-250ML, MERCK, Rahway, NJ, USA) gradient. After that, viral preparations were concentrated using Amicon Ultra Centrifugal Filters-Ultracel 100 K (#UFC910008. Millipore, Burlington, MA, USA) in a total volume of 1 mL. Titration of viral particles was performed by qPCR using primers complementary to the AAT promoter region: forward primer, 5′-CCCTGTTTGCTCCTCCGATA-3′, and reverse primer, 5′-GTCCGTATTTAAGCAGTGGATCCA-3′.

### 4.3. Animals and Animal Procedures

Animal procedures were conducted at CIMA Universidad de Navarra. All animal experiments were performed in strict accordance with the Animal Ethical Committee guidelines of the University of Navarra, and every effort was made to minimise the number of animals used and their suffering.

C57BL/6J mice were purchased from Jackson Laboratory (#000664) and bred at CIMA. Mice were group housed in ventilated cages (maximum of 6 animals per cage) in a temperature-controlled room and were kept on a 12 h light–dark cycle until the end of the study. Food and water were provided ad libitum. Treatments with rAAV vectors were performed in neonates and infantile mice (1, 2, 3, or 4 weeks of age) by intravenous injection (retro-orbital sinus) and with an rAAV dose of 5 × 10^12^ vg/kg. Groups were composed of 6–8 animals per group. Mice were randomly distributed in the different experimental groups. One, three, or six weeks after treatment, liver samples were collected from euthanised mice for histological analysis and nucleic acid extraction.

### 4.4. Molecular Analysis

#### 4.4.1. Nucleic Acids Extraction

For gDNA extraction, the NucleoSpin Tissue Kit (#740952.250, Macherey Nagel, D üren, Germany) was used following the provider’s instructions. RNA extraction was performed using trizol-chloroform by adding 200 μL of chloroform per ml of trizol. Both gDNA and RNA were quantified after extraction using the Nanodrop 1000 spectrophotometer (Thermo Scientific).

#### 4.4.2. qPCR

For viral genome quantification, primers were used for the detection of AAT (forward primer: 5′-CCCTGTTTGCTCCTCCGATA-3′ and reverse primer: 5′-GTCCGTATTTAAGCAGTGGATCCA-3′) and mGAPDH (forward primer: 5′-TGCACCACCAACTGCTTA-3′ and reverse primer: 5′-GGATGCAGGGATGATGTTC-3′) that was used as a normalising gene. qPCR was performed in a CFX96 Connect Real-Time Detection System (BioRad, Hercules, CA, USA) following this protocol: 95 °C for 10 min followed by 40 cycles of 95 °C for 15 s, 60 °C for 1 min, 72 °C for 25 s, and 80 °C for 10 s, and then, 95 °C for 1 min, 65 °C for 1 min, and a melting curve increase from 70 °C to 99.5 °C every 5 s.

#### 4.4.3. RT-qPCR

For transgene expression level measurement, 1 μg of RNA was reverse-transcribed into complementary DNA (cDNA) using M-MLV reverse transcriptase (#28025013, Invitrogen, Waltham, MA, USA). After that, specific primers for eGFP were used (forward: 5′-ATGGTGAGCAAGGGCGAGGA-3′ and reverse: 5′-TTGCCGGTGGTGCAGATGAA-3′) and for histone (forward: 5′-CCGCATACGTGGAGAACGTG-3′ and reverse: 5′-TCCCCTATTTTTCCACTCGCAA-3′), which was used as a normalising gene. RT-qPCR was performed in a CFX96 Connect Real-Time Detection System (BioRad) following this protocol: 95 °C for 3 min followed by 40 cycles of 95 °C for 15 s, 60 °C for 15 s, 72 °C for 25 s, 72 °C for 10 s, and 80 °C for 10 s, and then, 95 °C for 1 min, 65 °C for 1 min, and a melting curve increase from 70 °C to 99.5 °C every 5 s.

#### 4.4.4. Oligo dT PCR

For the assessment of mRNA stability, an oligo dT qPCR was performed. For this, during retro-transcription, oligo DT was added to the reaction and the Super Script IV First Strand Synthesis System (ThermoFisher Scientific) was used. RT-qPCR was performed in a CFX96 Connect Real-Time Detection System (BioRad) following this protocol: 95 °C for 3 min followed by 40 cycles of 95 °C for 15 s, 60 °C for 15 s, 72 °C for 25 s, 72 °C for 10 s, and 80 °C for 10 s, and then, 95 °C for 1 min, 65 °C for 1 min, and a melting curve increase from 70 °C to 99.5 °C every 5 s.

#### 4.4.5. Protein Extraction and Western Blot Analysis

Protein extractions from liver tissue samples were performed using 2% SDS buffer (Tris HCl 10 mM pH 7.4, SDS 2%, PMSF 1 mM, Aprotinin 1 ug/mL, Orthovanadate 1 mM, and Pyrophosphate 1 mM). Protein concentration was calculated using a BCA protein assay kit (#PI23225, ThermoFisher Scientific). Protein samples were then separated by SDS polyacrylamide gel electrophoresis (SDS-PAGE) and later transferred onto nitrocellulose trans-blot membranes (#1620115, BioRad). Nonspecific interactions of the antibodies were avoided by blocking the membranes with 5% non-fat milk for 30 min at RT. After the blocking, membranes were incubated with a primary antibody overnight at 4 °C. The following primary antibodies and conditions were used: α-GFP (1:2000; Abcam, Cambridge, UK, #AB6556) and α-β-Actin (1:5000; #A2228, SIGMA, St. Louis, MO, USA). After three washing steps with TBS-Tween 0.01%, the membranes were incubated with their respective secondary antibodies (α-Mouse IgG, HRP-linked Antibody #7076, Cell Signalling, Danvers, MA, USA, dilution 1:5000 for α-β-Actin; α-Rabbit IgG, HRP-linked Antibody, #7074 Cell Signalling, dilution 1:5000 for α-GFP) for 1 h at RT. Finally, chemiluminescence detection was performed using the ECL Ultra detection system (#TMA-100, Lumigen, Southfield, MI, USA) and an Odyssey^®^ Fc Imaging System (LI-COR). Signal quantification was performed using the ImageJ software (version 1.54).

### 4.5. IHC

Immunohistochemistry for GFP protein detection was performed in formalin-fixed, paraffin-embedded, 2 mm thick liver sections. α-GFP polyclonal antibody (1:2000; Abcam, #AB6556) was used for the staining of the different sections. Liver sections were also stained with haematoxylin and eosin (H&E) for routine histology analysis. For GFP+ area quantification, arbitrary images from each IHC-stained liver section were acquired. Images were analysed with a FIJI V1.46b plugin (ImageJ) developed by the Imaging Core Facility (CIMA, Universidad de Navarra, Pamplona, Spain) with which GFP expression in the liver was quantified. The quantification was expressed as the percentage of GFP-positive tissue area.

### 4.6. Statistical Analysis

All statistical analyses were performed using GraphPad Prism 7.0 for Windows. Significant differences between independent groups were analysed using a one-way ANOVA followed by a Tukey’s multiple comparisons post hoc test, a Mann–Whitney test, or a mixed-effects model analysis. *p* < 0.05 (*) was considered significant. All graphs show the mean values ± SEM.

## 5. Patents

The patent “GENE THERAPY VECTORS COMPRISING S/MAR SEQUENCES” with the publication number: WO/2019/219649 resulted from this work.

## Figures and Tables

**Figure 1 ijms-25-12734-f001:**
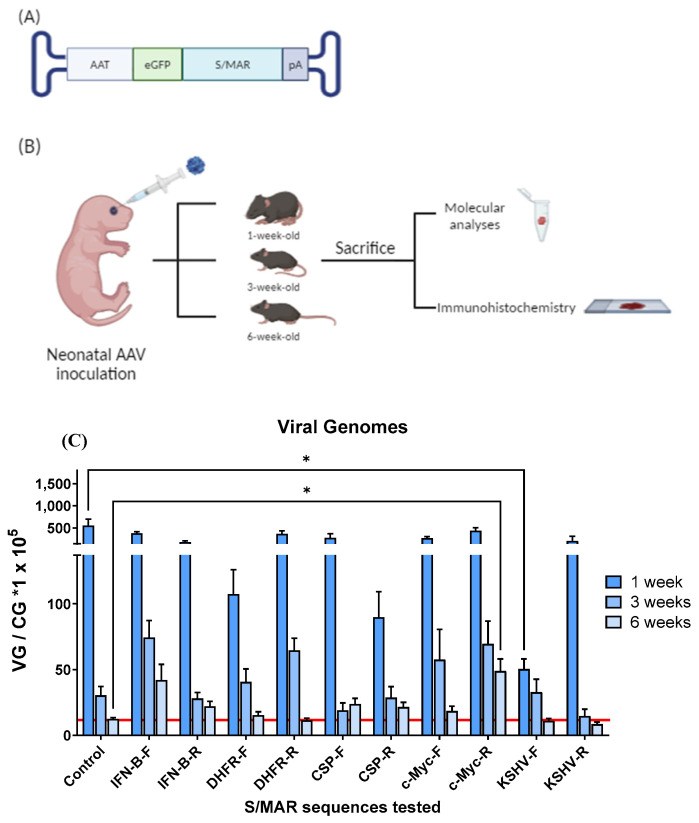
S/MAR candidate selection: (**A**) schematic representation of the rAAVs injected and (**B**) the experimental workflow (Created in BioRender. Llanos ardaiz, A. (2024) https://BioRender.com/s33k203, accessed on 23 November 2024) followed for the selection of the S/MAR candidate in neonate C57BL/6 mice. (**C**) Analysis of viral genomes (VG) in the livers of mice treated with the rAAVs containing the different S/MAR sequences tested. rAAV viral genome levels were measured by qPCR 1, 3, or 6 weeks post-neonatal injection of C57BL/6 mice with a dose of 5 × 10^12^ vg/kg. Total DNA values were normalised to *GAPDH* levels (values *1 × 10^5^). The red line indicates viral genome levels in the control group 6 weeks post-injection. CG: cell genome. Data are represented as mean + SEM. * indicates *p*-value 0.01 to 0.05 (mixed-effects analysis).

**Figure 2 ijms-25-12734-f002:**
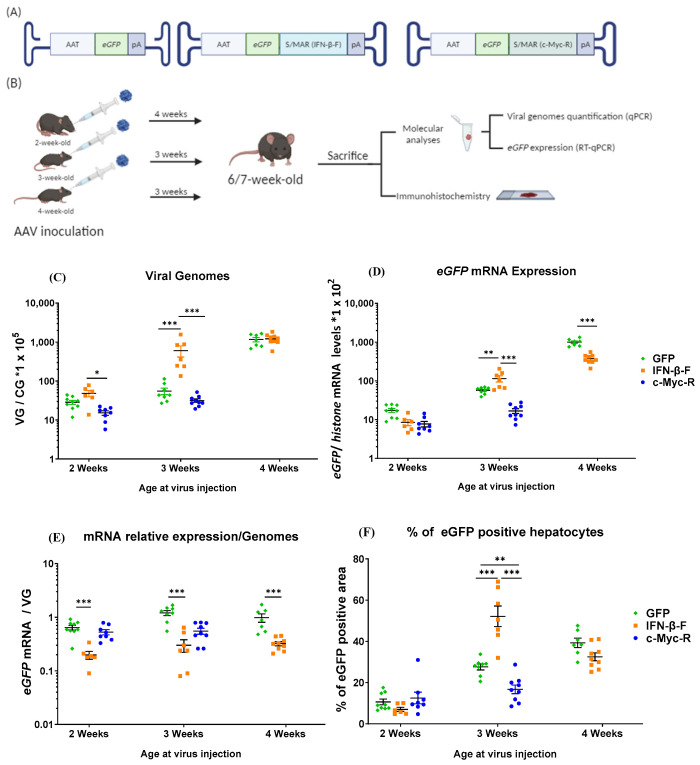
Inoculation age selection: (**A**) schematic representation of the rAAVs injected and (**B**) the experimental workflow (Created in BioRender. Llanos ardaiz, A. (2024) https://BioRender.com/n18x984, accessed on 23 November 2024). Viral genomes (**C**) and transgene expression (**D**) were analysed in the livers of mice treated at 2, 3, or 4 weeks of age with either a control vector or the vectors containing the IFN-β-F or c-Myc-R S/MAR sequences. When the mice reached 6 or 7 weeks of age, rAAV viral genome levels were measured by qPCR with total DNA values normalised to *GAPDH* levels (values *1 × 10^5^). *eGFP* expression levels were measured by RT-qPCR with total RNA levels normalised to *histone* mRNA levels (values *1 × 10^2^). The ratio of the mRNA relative expression per viral genome was calculated (**E**) and the percentage of eGFP positive hepatocytes (**F**) was quantified by immunohistochemistry analysis. Values are represented as percentage of eGFP positive area over total area. CG: cell genome. Data are represented as mean ± SEM. * indicates *p*-value 0.01 to 0.05, ** indicates *p*-value 0.001 to 0.01, *** indicates *p*-value 0.0001 to 0.001 (Mann–Whitney test analysis).

**Figure 3 ijms-25-12734-f003:**
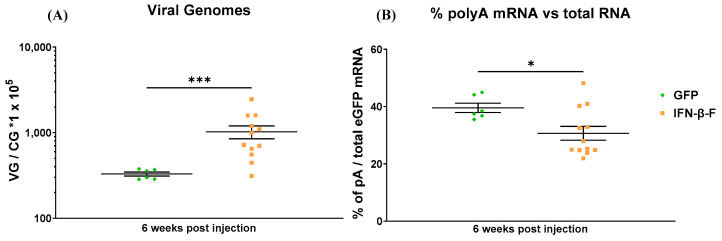
mRNA quality of S/MAR-containing vectors: (**A**) viral genomes and (**B**) mRNA stability study in samples from mice treated with either the control vector or the vector containing the IFN-β-F S/MAR sequence at 3 weeks old. Livers were collected 6 weeks post viral injection. DNA and RNA were extracted. Viral genome copies were quantified by qPCR. For RNA stability assay, two different retro-transcription protocols were followed, one using the MLV reverse transcriptase and random primers and another using the Super Script IV First Strand Synthesis System and oligo DT. To assess RNA stability, RNA was retro-transcribed using both methods, and then *eGFP* expression was quantified by RT-qPCR. The percentage of polyA-containing (pA) mRNA over total RNA was calculated. CG: cell genome. Data are represented as mean + SEM. * indicates *p*-value 0.01 to 0.05, *** indicates *p*-value 0.0001 to 0.001 (Mann–Whitney test).

**Figure 4 ijms-25-12734-f004:**
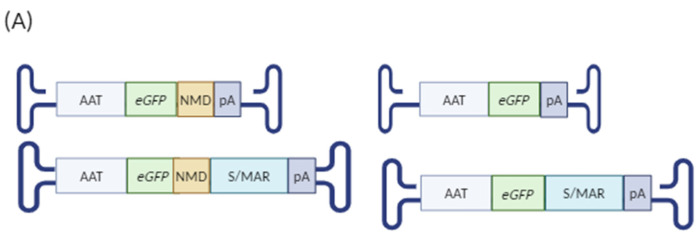
Evaluation of NMD-inhibitory sequences: (**A**) schematic representation of the rAAVs injected and (**B**) the experimental workflow followed (Created in BioRender. Llanos ardaiz, A. (2024) https://BioRender.com/w27v327, accessed on 23 November 2024). Analysis of viral genomes (**C**), transgene expression (**D**), and *eGFP* expression per rAAV vector genome (VG) (**E**) in the livers of mice treated at 3 weeks old with either a control vector or the vectors containing the NMD-inhibitory sequences with or without the IFN-β-F S/MAR. rAAV viral genome levels were measured by qPCR with total DNA values normalised to *GAPDH* levels (values *1 × 10^5^). *eGFP* expression levels were measured by RT-qPCR with total RNA levels normalised to *histone* levels (values *1 × 10^2^). CG: cell genome, WB: Western blot. Data are represented as mean + SEM. * indicates *p*-value 0.01 to 0.05, ** indicates *p*-value 0.001 to 0.01, and **** indicates *p*-value < 0.0001 (ordinary one-way ANOVA).

**Figure 5 ijms-25-12734-f005:**
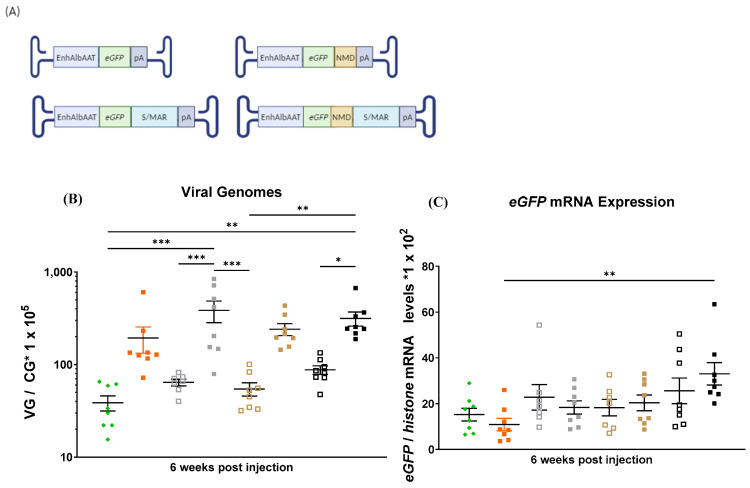
Evaluation of NMD-inhibitory sequences under the EnhAlbAAT promoter: schematic representation of vectors used (Created in BioRender. Llanos ardaiz, A. (2024) https://BioRender.com/a68c078, accessed on 23 November 2024) (**A**), analysis of viral genomes (**B**), transgene expression (**C**), *eGFP* expression per rAAV genome (**D**), eGFP protein quantification (**E**), and percentage of positive hepatocytes (**F**) in the livers of mice treated at 3 weeks of age with either a control vector or the vectors containing the NMD-inhibitory sequences with or without the IFN-β-F S/MAR sequence and under the EnhAlbAAT promoter and sacrificed 6 weeks later. rAAV viral genomes levels were measured by qPCR with total DNA values normalised to *GAPDH* levels (values *1 × 10^5^). *eGFP* expression levels were measured by RT-qPCR with total RNA levels normalised to *histone* levels (values *1 × 10^2^). eGFP protein expression was quantified by WB with normalisation to B-Actin protein levels. Percentage of positive hepatocytes was quantified from IHC by quantifying the eGFP positive area over the total area. CG: cell genome. Data are represented as mean ± SEM. * indicates *p*-value 0.01 to 0.05, ** indicates *p*-value 0.001 to 0.01, *** indicates *p*-value 0.0001 to 0.001, and **** indicates *p*-value < 0.0001 (ordinary one-way ANOVA).

## Data Availability

The original contributions presented in the study are included in the article material; further inquiries can be directed to the corresponding authors.

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
