# Peer review of "In Vivo Selection of S/MAR Sequences to Favour AAV Episomal Maintenance in Dividing Cells"

_ijms, 2024, doi:10.3390/ijms252312734_

Round 1

Reviewer 1 Report

Comments and Suggestions for Authors

In this MS the authors aim to improve AAV vector stability and expression in hepatocytes of infantile mice. They tested different S/AR region previously identified and cloned and different NMD inhibitory sequence previously reported .

The study is well-structured and the results were clearly described in the text. Unfortunately, all figures were not correctly included in the text. However, as claimed by the authors, the results reported in this study still need to be fully elucidated before proposing a new design of AAV vector including S/MAR with /without NMD inhibitory elements for therapeutic purposes.

To the reviewer’s opinion, addressing the following points would enhance clarity and further strengthen the manuscript.

-After screening of 5 S/MAR regions in for and rev orientation, the authors identified IFN beta f S/MAR able to improve the amount of AAV viral genome still present in the liver of infantile mice 3 wks post injection (p.i.) (fig .2). This effect was not observed at 4 week p.i . Why? Please, Discuss . moreover, the fig2 lacks the data on cmyc vector 4 wks post injection. Please show the data in the figure 2D.

-The IFN beta f S/MAR negatively impacts the GFP expression carried by the AAV . the authors suggests that this un expected effect could be due to the episomal state of the AAV . Could the authors extend the discussion to an AAV carrying a cDNA of a potential therapeutic gene which could include also part of the 3’ UTR region and to Adenovector  (episomal vector)including also S/MAR region?

-To ameliorate the GFP expression they tested 3 NMD inhibitory elements cloned into the AAV S/MAR vector. Three wks post injection they observed that all NMD inhibitory elements interfere with the S/MAR element's role in rAAV genome maintenance. Replacing the AAT promoter with a stronger enhancer alb+ AAT  promoter, all NMD inhibitory elements benefit the AAV genome stability in presence of S/MAR, but only 1 NMD inhibitory element(hTMED) significantly increased eGFP protein levels, regardless of the presence of the S/MAR element in the vector. (Fig5E). However the % of GFP  hepatocyte were comparable among all vectors regardless the presence of S/MAR or NMD inhibitory sequence indicating. Could the authors better discuss the effect of hTEMD and why this vector is not statistically significant in. fig 5E?

-Fig3: the authors showed in fig 2d the IFN-β-F S/MAR  AAV vector express more GFP than GFP vector. In fig 3 they show that the mRNA   IFN-β-F S/MAR  AAV  is less stable. Could they clearly explain? Since the authors did not performed an analysis of RNA quality by Bioanalyzer, What about the quality of the extracted RNA? Moreover could the authors include the data from mice injected at 2 and 4 week ?

-Fig 5 EnhAlbAAT-hTMED vs EnhAlbAAT-hTMED-S/MAR mRNA and protein level are Comparable?

Minor issues:

Line 514 indicate cat number of thermos fisher plasmid

Line 516 add references of the pERI-IR plasmids

Line 565 add catalog number

Line 625 state number of mice injected  per group

Line 658-659 typo oligo dT

Line 263 specfify which S/MAR was selected

Author Response

Comments and Suggestions for Authors

In this MS the authors aim to improve AAV vector stability and expression in hepatocytes of infantile mice. They tested different S/AR region previously identified and cloned and different NMD inhibitory sequence previously reported.

The study is well-structured, and the results were clearly described in the text. Unfortunately, all figures were not correctly included in the text. However, as claimed by the authors, the results reported in this study still need to be fully elucidated before proposing a new design of AAV vector including S/MAR with /without NMD inhibitory elements for therapeutic purposes.

To the reviewer’s opinion, addressing the following points would enhance clarity and further strengthen the manuscript.

First of all thank you very much to the reviewer for the insightful comments and help to improve the manuscript.

-After screening of 5 S/MAR regions in for and rev orientation, the authors identified IFN beta f S/MAR able to improve the amount of AAV viral genome still present in the liver of infantile mice 3 wks post injection (p.i.) (fig .2). This effect was not observed at 4 week p.i . Why? Please, Discuss . moreover, the fig2 lacks the data on cmyc vector 4 wks post injection. Please show the data in the figure 2D.

We acknowledge that the ages at which mice were injected and sacrificed are confusing and therefore would like to clarify that there were three cohorts of animals that differ in the age of treatment: 2 weeks (cohort A), 3 weeks (cohort B) and 4 weeks of age (cohort C). Cohort A was sacrificed 4 weeks p.i. while cohorts B and C were sacrificed 3 weeks p.i. No mice from cohort C (inoculated at 4 weeks of age) were injected with the c-Myc-R vector. Results from the c-Myc-R construct were obtained first for cohorts A and B and due to the mediocrity of the data, mice from cohort C were not injected with this vector.

The reason for the lack of any benefit conveyed by the presence of S/MAR when injected at 4 weeks of age is related to the mechanism of action of the S/MAR sequence and the proliferative status of hepatocytes. While at 3 weeks of age hepatocytes are still undergoing proliferation, at 4 weeks the proliferation rate is negligible (Septer S, et al. Am J Physiol Gastrointest Liver Physiol. 2012 Mar 1;302(5):G493-503.). The S/MAR sequence reduces the extent of loss of AAV genomes in dividing hepatocytes, and for this reason significant differences are observed in 3-week-old animals but not in 4-week-old mice.

-The IFN beta f S/MAR negatively impacts the GFP expression carried by the AAV. the authors suggests that this unexpected effect could be due to the episomal state of the AAV. Could the authors extend the discussion to an AAV carrying a cDNA of a potential therapeutic gene which could include also part of the 3’ UTR region and to Adenovector (episomal vector) including also S/MAR region?

-As suggested in the manuscript, we believe that the negative impact of the S/MAR sequence in eGFP expression is due, at least in part, to the increased length of the 3’UTR of the resulting mRNA which, as a consequence, is more unstable and susceptible to RNA degradation (Figure 3B). We have preliminary data generated in a disease mouse model in which we have evaluated an AAV vector carrying a therapeutic gene and the S/MAR sequence selected in the present manuscript. Results from this study suggest that the presence of the S/MAR sequence negatively impacts the expression of the therapeutic transgene in a similar way to what we have observed here with eGFP. However, the therapeutic effect of this vector is still superior to the one not carrying the S/MAR sequence, due to an increased maintenance of AAV genomes in a higher number of cells.

The inclusion of the S/MAR sequence could benefit the maintenance in dividing cells of other episomal vectors such as third generation or high-capacity Adenoviral vectors; however, it will not improve the maintenance of first-generation adenovirus, since the short-term expression associated with these vectors is due to the activation of innate and adaptive immune responses against the vector. Other groups have performed experiments introducing the S/MAR sequence into other episomal vectors like plasmids or non-integrative lentiviral vectors obtaining results that are in line with our results (Bode & Maass, 1988) (Schübeler et al., 1996)(Verghese et al., 2014).

-To ameliorate the GFP expression they tested 3 NMD inhibitory elements cloned into the AAV S/MAR vector. Three wks post injection they observed that all NMD inhibitory elements interfere with the S/MAR element's role in rAAV genome maintenance. Replacing the AAT promoter with a stronger enhancer alb+ AAT promoter, all NMD inhibitory elements benefit the AAV genome stability in presence of S/MAR, but only 1 NMD inhibitory element(hTMED) significantly increased eGFP protein levels, regardless of the presence of the S/MAR element in the vector. (Fig5E). However, the % of GFP hepatocyte were comparable among all vectors regardless the presence of S/MAR or NMD inhibitory sequence indicating. Could the authors better discuss the effect of hTEMD and why this vector is not statistically significant in. fig 5E?

The hTEMD element has shown to improve eGFP mRNA levels and protein (measured by Western blot) as represented in Figure5 C and E, respectively. However, this improvement in comparison with the control vector (with or without S/MAR) was not reproduced when % of eGFP positive area was assessed. This might be due to the fact that the techniques have very important differences, e.g., the analysis of total mRNA or total protein doesn’t take into account the number of cells that are transduced, while the determination of the number of eGFP positive cells identifies the number of cells that are transduced but does not indicate the amount of protein per cell. Thus, the data will indicate that the percentage of transduce cells are similar but the addition of the hTEMD element increases the levels of mRNA and protein essentially by acting as an enhancer.

- As for the absence of statistical significance mentioned by the reviewer, when performing a One-Way ANOVA with the Tukey correction for multiple comparisons there is no significant difference between the EnhAlbAAT-GFP vector and the EnhAlbAAT-GFP-hTMED-SMAR vectors. This was the specific statistical test applied to the data represented in figure 5. However, using a Newman-Keuls correction, which holds less validity for statistical comparison in this instance, or not correcting for multiple comparisons, statistical significances appear between these two groups. We wanted to be as conservative as we could to maintain the balance between statistical power and type I error rate (false positive error rate)(Lee & Lee, 2018).

-Fig3: the authors showed in fig 2d the IFN-β-F S/MAR AAV vector express more GFP than GFP vector. In fig 3 they show that the mRNA   IFN-β-F S/MAR AAV is less stable. Could they clearly explain? Since the authors did not perform an analysis of RNA quality by Bioanalyzer, What about the quality of the extracted RNA? Moreover, could the authors include the data from mice injected at 2 and 4 week?

-It is true that in Figure 2D the levels of eGFP mRNA expression are higher in the IFN-β-F S/MAR group than in the control group and that in Figure 3 the mRNA from this vector has lower stability than the control vector. It is important to consider that the mRNA stability analysis was not performed with the same samples as those used for the RT-qPCR performed to assess the levels of eGFP mRNA from figure 2D. While the results from figure 2D to which the reviewer refers were obtained in mice injected at 3 weeks of age and sacrificed 3 weeks p.i., the results from figure 3 were obtained from mice injected at 3 weeks of age and sacrificed 6 weeks p.i. This may explain the discrepancy between the levels of transgene expression and mRNA quality, due to the different time points analysed.

-Regarding the quality of the extracted RNA, quantification of extracted RNA was performed using a Nanodrop 1000 spectrophotometer (Thermo Scientific). Ratios of 260/280 absorbance of the different samples, independently of the vector injected, are ~2.0, which usually indicates pure RNA.

The quality of the mRNA coming from mice injected at 2 and 4 weeks, was also assessed in this way.

-Fig 5 EnhAlbAAT-hTMED vs EnhAlbAAT-hTMED-S/MAR mRNA and protein level are Comparable?

-The levels of mRNA and protein are similar in both groups, although a trend is observed that shows that with the inclusion of the S/MAR there is a slight increase in the mRNA levels. However, the increase in mRNA is not proportional to the increase in the genome levels for the vector containing the S/MAR sequence, indicating once again that although the SMAR sequence has a positive effect over vector genome maintenance it has a negative impact on transgene expression levels.

Minor issues:

Line 514 indicate cat number of thermos fisher plasmid

These products were custom orders specifically generated for this project, and thus there is no catalogue number. This information is now in line 552.

Line 516 add references of the pERI-IR plasmids

References have been added. This information is now in line 553.

Line 565 add catalog number

These products were custom orders specifically generated for this project, and thus there is no catalogue number. This information is now in line 603.

Line 625 state number of mice injected per group

Done. This information is now in lines 664-665.

Line 658-659 typo oligo dT

Done. This information is now in lines 696-697.

Line 263 specify which S/MAR was selected

Done. This information is now in line 265.

Reviewer 2 Report

Comments and Suggestions for Authors

The paper by Llanos-Ardaiz and colleagues investigates possible AAV vector genome designs that promote the long-term maintenance of viral episomes in cells with high replication rates, in which the vector is rapidly diluted. This feature would be particularly useful in case of gene therapy treatments in infancy in still growing organs such the liver.

The experiments presented in the manuscript evaluate the stability effects of S/MAR and NMD elements, known to either anchor episomes to host cell chromatin or protect transcribed RNA from degradation, when inserted into an AAV backbone, alone or in combination, in a mouse model of liver gene transfer.

Overall, the obtained results indicate that a strategy aimed at improving vector genome maintenance will require a careful selection and tuning of regulatory elements, including the type and strength of promoters. Moreover, also the choice of an optimal age for gene transfer, ideally targeting a stage with moderately proliferating liver cells, seems to be crucial for long term preservation of AAV genomes.

The experiments are well executed and clearly presented, with critical discussion of the results. While this work does not achieve any groundbreaking breakthroughs, it addresses a challenging topic and establishes important foundations for future, more impactful research.

Comments.

i) Even with the more effective S/MAR element and in the presence of the NMD element, there is a consistent decline in viral genomes and GFP expression. Could this modest vector improvement hold any practical therapeutic value?

ii) One limitation noted is that the presence of regulatory elements in the AAV vector backbone significantly reduced the already limited cloning capacity of AAV vectors. The authors correctly highlight this issue. Do they observe any problems with vector production? Does the presence of S/MAR and NMD elements affect the total yield or final titer of AAV preparations?

iii) The sharp drop in genome levels observed in transduced mice during the first weeks of life may suggest the involvement of mechanisms beyond simple dilution due to cell division. Have the authors considered the possibility of AAV genome degradation by the DNA repair machinery, which is particularly active in highly proliferative cells?

Author Response

i) Even with the more effective S/MAR element and in the presence of the NMD element, there is a consistent decline in viral genomes and GFP expression. Could this modest vector improvement hold any practical therapeutic value?

We believe that the 5-fold increase in vector genomes observed in mice injected at 3-weeks of age after the inclusion of the S/MAR and the NMD elements into the eGFP-expressing vector could have therapeutic value. We modestly think that the increase in viral genomes is quite remarkable due to the fact that livers are constantly undergoing cell division throughout the duration of the studies. We especially think that it could have value in treating our target population which are infants with rare inherited liver disorders, whose treatment with AAV-based gene therapy vectors is normally restricted due to the fact that the therapeutic vector will get diluted during liver growth. We agree that the results may be modest, but we believe that this is a promising starting point that requires optimisation and probably fine tuning of all elements in the construct, for the generation of therapeutic vectors for the treatment of infants with growing organs. Moreover, previous studies have shown that even only a slight increase in the number of rAAV genomes and transgene expression can differentiate a vector from being therapeutic or not (Weber et al., 2019), supporting our belief that the inclusion of these sequences, that results in an increase in viral genomes, could have therapeutic value.

ii)One limitation noted is that the presence of regulatory elements in the AAV vector backbone significantly reduced the already limited cloning capacity of AAV vectors. The authors correctly highlight this issue. Do they observe any problems with vector production? Does the presence of S/MAR and NMD elements affect the total yield or final titer of AAV preparations?

No problems were encountered in AAV vector production despite the large size of the genome. Titers and production quality (SYPRO staining of capsid proteins) were checked in all AAV vector preparations. No striking differences were observed neither in titers nor in the quality of the vectors generated except for vector  ss pAAV- AAT eGFP RSV-RSE  pA (info attached).

iii) The sharp drop in genome levels observed in transduced mice during the first weeks of life may suggest the involvement of mechanisms beyond simple dilution due to cell division. Have the authors considered the possibility of AAV genome degradation by the DNA repair machinery, which is particularly active in highly proliferative cells?

Thanks to the reviewer for this insightful comment. The reviewer is right that the DNA repair machinery could be involved in the dramatic loss of AAV genomes. As previously described by Mondal et al., 2023, “fetal and neonatal livers are comprised of proliferative hepatocytes with abundant expression of genes involved in homology-directed repair (HDR)…”. Some studies have shown that ssAAV genomes might be degraded rapidly as a signal of DNA damage (Hauck et al., 2004), which could partially explain the rapid decrease in viral genomes that we observe. Also, the stabilization of episomal AAVs requires host DNA repair mechanisms (Bijlani et al., 2021). Thus, the rapid loss of AAV genomes in neonates might be related to an increased activity of the DNA repair machinery in addition to the vector dilution due to the high proliferation rate of hepatocytes (this point has been now discussed).

Reviewer 3 Report

Comments and Suggestions for Authors

Title: In vivo selection of S/MAR sequences to favour AAV episomal maintenance in dividing cells

·       Summary

In this manuscript, the authors present a study on the testing of S/MAR sequences to enhance AAV episomal maintenance in dividing cells. This research is both valuable and critical for the selection of S/MAR sequences. The authors have organized their study effectively and have written the manuscript clearly. However, I recommend that the authors carefully consider the following questions during their review.

·       Major issues

1.      All figures lack labels for the X-axis and Y-axis. Please ensure these are included.

2.      The figures sourced from BioRender.com do not include proper citations. Please review BioRender.com's citation policy.

3.      In Figure 2A, the S/MAR box is missing. Please verify this.

4.      In Figures 1C, 2C, 3A, 4C, and 5B, the title is labeled as "Viral Genomes." However, the figure legends refer to it as "liver transduction." While the viral genomes were analyzed in liver samples, the titles in the figure legends should be reviewed for consistency.

·       Major issues

1.      In line 186, to make uniform, IFN-β-For à IFN-β-F

2.      In line 186, to make uniform, c-Myc-Rev à c-Myc-R

3.      In line 188, to make uniform, 10E5 à 105

4.      In line 188, RTqPCR à RT-qPCR

5.      In line 192, + SEM à +- SEM

6.      In line 203, cMyc-R à c-Myc-R

7.      In line 215, ORF à open reading frames (ORF)

8.      In line 372, 3weeks à 3 weeks

9.      In line 376, 646, RTqPCR à RT-qPCR

10.  In line 379, + SEM à +- SEM

11.  In line 503, 552, 554, SMAR à S/MAR

Author Response

Major issues

  1. All figures lack labels for the X-axis and Y-axis. Please ensure these are included.

Corrected. Labels for the X and Y axes were included.

2.The figures sourced from BioRender.com do not include proper citations. Please review BioRender.com's citation policy.

Proper citations have been added according to BioRender.com’s citation policy.

3.In Figure 2A, the S/MAR box is missing. Please verify this.

Corrected. Boxes “IFN-B-F” and “c-Myc-R “were renamed as: “S/MAR (IFN-B-F)” and “S/MAR (c-Myc-R)”.

4.In Figures 1C, 2C, 3A, 4C, and 5B, the title is labeled as "Viral Genomes." However, the figure legends refer to it as "liver transduction." While the viral genomes were analyzed in liver samples, the titles in the figure legends should be reviewed for consistency.

This has been addressed with all the legends referring to “viral genomes”.

Major issues

  1. In line 186, to make uniform, IFN-β-For-aIFN-β-F

All instances of IFN-β-For have been changed to IFN-β-F. This information is now in line 187.

  1. In line 186, to make uniform, c-Myc-Rev-c-Myc-R

All instances of c-Myc-Rev have been changed to c-Myc-R. This information is now in line 187.

  1. In line 188, to make uniform, 10E5 5x105

Corrected.  This information is now in line 189.

  1. In line 188, RTqPCR à RT-qPCR

Corrected. This information is now in line 189.

  1. In line 192, + SEM+- SEM

Corrected. This information is now in line 193.

  1. In line 203, cMyc-Rc- Myc-R

All instances of cMyc-Rev have been changed to c-Myc-R. This information is now in line 204.

  1. In line 215, ORF à open reading frames (ORF)

This acronym was defined earlier (line 103).

  1. In line 372, 3weeks à 3 weeks

Corrected. This information is now in line 402.

  1. In line 376, 646, RTqPCR à RT-qPCR

Corrected. This information is now in lines 406 and 684.

  1. In line 379, + SEM+- SEM

Corrected. This information is now in line 409

  1. In line 503, 552, 554, SMAR S/MAR

Corrected. This information is now in lines 540, 559 and 561.

Round 2

Reviewer 1 Report

Comments and Suggestions for Authors

Authors: We acknowledge that the ages at which mice were injected and sacrificed are confusing and therefore would like to clarify that there were three cohorts of animals that differ in the age of treatment: 2 weeks (cohort A), 3 weeks (cohort B) and 4 weeks of age (cohort C). Cohort A was sacrificed 4 weeks p.i. while cohorts B and C were sacrificed 3 weeks p.i. No mice from cohort C (inoculated at 4 weeks of age) were injected with the c-Myc-R vector. Results from the c-Myc-R construct were obtained first for cohorts A and B and due to the mediocrity of the data, mice from cohort C were not injected with this vector.

Reviewer: thanks for your answer. The scheme of the experiment was clear to the reviewer, but please specify in the text, results, line 155, that mice at 4wks of age were not injected with c-myc vector.

authors: The S/MAR sequence reduces the extent of loss of AAV genomes in dividing hepatocytes, and for this reason significant differences are observed in 3-week-old animals but not in 4-week-old mice.

reviewer: the authors stated no (negligible) proliferation of hepatocyte after 3wks of age, indeed viral genomes in fig 2c are comparable (see 4wks injected mice) . in mice injected at 3 wks of age and analysed 6 wks later viral genome increased in presence of S/MAR (Fig 3A). Does it  means that  just a week- window (between 3 and 4 wks old mice) is crucial to explain the role of the S/MAR  in increasing the persistence of the viral genome? Please include in the discussion.

Could you cite figure 3A in the text (results)?

authors :Regarding the quality of the extracted RNA, quantification of extracted RNA was performed using a Nanodrop 1000 spectrophotometer (Thermo Scientific). Ratios of 260/280 absorbance of the different samples, independently of the vector injected, are ~2.0, which usually indicates pure RNA.

reviewer : the nanodrop analysis is really a raw quality control for RNA.  However, since data from fig2 are not comparable with data in fig 3, the quality of mRNA is no more an issue.

authors These products were custom orders specifically generated for this project, and thus there is no catalogue number. This information is now in line 552.

Reviewer: Info still lacking. Please specify custom orders.  

authors products were custom orders specifically generated for this project, and thus there is no catalogue number. This information is now in line 603. Specify custom made

Reviewer Info still lacking. Please specify custom orders

Author Response

Authors: We acknowledge that the ages at which mice were injected and sacrificed are confusing and therefore would like to clarify that there were three cohorts of animals that differ in the age of treatment: 2 weeks (cohort A), 3 weeks (cohort B) and 4 weeks of age (cohort C). Cohort A was sacrificed 4 weeks p.i. while cohorts B and C were sacrificed 3 weeks p.i. No mice from cohort C (inoculated at 4 weeks of age) were injected with the c-Myc-R vector. Results from the c-Myc-R construct were obtained first for cohorts A and B and due to the mediocrity of the data, mice from cohort C were not injected with this vector.

Reviewer: thanks for your answer. The scheme of the experiment was clear to the reviewer, but please specify in the text, results, line 155, that mice at 4wks of age were not injected with c-myc vector.

-As kindly suggested by the reviewer, we have included an explanatory sentence in the text to hopefully make this point clearer for the reader.

authors: The S/MAR sequence reduces the extent of loss of AAV genomes in dividing hepatocytes, and for this reason significant differences are observed in 3-week-old animals but not in 4-week-old mice.

reviewer: the authors stated no (negligible) proliferation of hepatocyte after 3wks of age, indeed viral genomes in fig 2c are comparable (see 4wks injected mice) . in mice injected at 3 wks of age and analysed 6 wks later viral genome increased in presence of S/MAR (Fig 3A). Does it  means that  just a week- window (between 3 and 4 wks old mice) is crucial to explain the role of the S/MAR  in increasing the persistence of the viral genome? Please include in the discussion.

The rate of hepatocyte proliferation and cell division significantly influences the effectiveness of the S/MAR sequence. The mechanism of action of the S/MAR element involves anchoring the DNA to the nuclear matrix, thereby facilitating DNA replication during cellular DNA synthesis and division [23, 24]. Consequently, the activity of the S/MAR sequence is closely tied to the replication status of the cells. In the absence of cell division, the S/MAR sequence does not confer any advantage. This is evident in the liver of 4-week-old animals, where hepatocyte proliferation has largely ceased. In contrast, in the liver of 3-week-old animals, hepatocytes are still dividing. We hypothesize that in these younger animals, some AAV vector genomes containing the S/MAR sequence are replicated together with the cellular DNA.

This hypothesis explains the observed advantage of incorporating the S/MAR element when vectors are administered at 3 weeks of age, compared to the lack of such benefit when administered at 4 weeks of age.

-This specific topic has been addressed in the discussion.

Could you cite figure 3A in the text (results)?

-Figure 3A has been referenced in the text.

authors :Regarding the quality of the extracted RNA, quantification of extracted RNA was performed using a Nanodrop 1000 spectrophotometer (Thermo Scientific). Ratios of 260/280 absorbance of the different samples, independently of the vector injected, are ~2.0, which usually indicates pure RNA.

reviewer : the nanodrop analysis is really a raw quality control for RNA.  However, since data from fig2 are not comparable with data in fig 3, the quality of mRNA is no more an issue.

-The reviewer is right, the nanodrop analysis is a raw quality control for RNA since it only provides data of absorbance and not of RNA sequence integrity. Therefore, our statements were based on the purity (nanodrop absorbance ratios) of the extracted RNA and the qPCR data for stability assessment.

 -We are glad that the quality of mRNA is no longer an issue

authors These products were custom orders specifically generated for this project, and thus there is no catalogue number. This information is now in line 552.

Reviewer: Info still lacking. Please specify custom orders.  

-We have specified in the text that these sequences were custom ordered.

authors products were custom orders specifically generated for this project, and thus there is no catalogue number. This information is now in line 603. Specify custom made

Reviewer Info still lacking. Please specify custom orders

-We have specified in the text that these sequences were custom ordered.